# Exposure assessment of elemental carbon, polycyclic aromatic hydrocarbons and crystalline silica at the underground excavation sites for top-down construction buildings

Hyunhee Park [1,2], Eunsong Hwang[1], Miyeon Jang[1], Chungsik Yoon [2]*

1 Work Environment Research Bureau, Occupational Safety and Health Research Institute, Ulsan, Republic of Korea, 2 Department of Occupational Health, Graduate School of Public Health, Seoul National University, Seoul, Republic of Korea

* csyoon@snu.ac.kr

**Data Availability Statement:** All relevant data are within the manuscript and its Supporting Information files.

## Abstract

Enclosed underground excavation worksite has an environment with poor ventilation and exposure to hazardous substances from diesel engine exhaust and construction materials. The objective of this study was to evaluate the exposure level of elemental carbon (EC), organic carbon (OC), total carbon (TC), polycyclic aromatic hydrocarbons (PAHs), dust and crystalline silica (CS) during underground excavation work for top down construction buildings. Active local air sampling for EC, OC, and TC (n = 105), PAHs (n = 50), dust (n = 34) and CS (n = 34) was conducted from inside and outside the excavator at underground excavation workshop in four different construction sites. EC, OC, TC and CS were sampled with each respirable and total particulates. EC, OC, and TC were collected on quartz-filter and analyzed using the thermal optical transmittance method. PAHs was collected on polytetra-fluorethylene filter with XAD-2 and analyzed using liquid chromatography with fluorescence detector. CS and particulates were collected on poly vinyl chloride filter and analyzed using fourier-transform infrared spectroscopy. The geometric mean of respirable EC, OC, TC, total PAHs, respirable dust and respirable CS were 8.69 μg/m³, 34.32 μg/m³, 44.96 μg/m³, 6.818 μg/m³ 0.13 mg/m³ and 0.02 mg/m³ from inside the excavator and 33.20 μg/m³, 46.53 μg/m³, 78.21 μg/m³, 3.934 μg/m³, 0.9 mg/m³ and 0.08 mg/m³ from outside the excavator (underground excavation workshop), respectively. The EC and RCS concentration from outside the excavator is significantly higher than that of inside the excavator (p<0.01). The worksite with rock ground, higher vehicle density, blasting and enclosed environments had higher exposure to EC than other sites (p<0.05). There was no significant difference of EC concentration between total and respirable particulates. In top down construction sites, EC concentrations during underground excavation work exceeded recommended exposure limits as 20 μg/m³, accounted for about 50% of the total sample, and the level of concentration of RCS exceeded 1.5 times of occupational exposure limit, 0.05 mg/m³. Efforts are needed to minimize exposure to diesel engine exhaust and silica in underground excavation

**Funding:** This work was supported by the Occupational Safety & Health Research Institute (OSHRI), Korea Occupational Safety & Health Agency(KOSHA) and funded by the research project(No. 2017-OSHRI-1061) of the OSHRI (URL: http://oshri.kosha.or.kr/eoshri/index.do). The funders had role in study design, data collection and analysis, decision to publish, or preparation of the manuscript.

**Competing interests:** The authors have declared that no competing interests exist.

sites. Management of diesel engine vehicle, supply of fresh air and ventilation and introducing water facilities to create wet environment in underground worksites are strongly suggested.

## Introduction

Crowded urban areas in Asia have experienced a recent increase in the number of construction sites that employ top-down excavation methods [1]. Top-down methods are those where the both super and sub-structures are simultaneously built, and they are useful in urban areas where there are strict environmental regulations, lack of working space, and short construction times [2]. Top-down construction methods include the installation of perimeter retaining walls, pre-founded columns and a horizontal structure for the support from the ground before initiating excavation work, and at this time, the floor slab is installed above the underground workplace as the excavation proceeds downwards [1, 3]. Since the underground workplace is enclosed, internal ventilation therein is very poor for the excavation work. The workers engaged in excavation are exposed to diesel engine exhaust (DEE) from excavators and trucks, dust emitted from rock excavation, respirable crystalline silica (RCS), and other fumes and particulate matter.

Previous studies have reported the concentration of RCS in construction sites based on typical occupations of tunnel construction worker [4–6], cement mason and concrete finisher [5–8], and building demolition [5, 9–11]. The concentration of DEE in construction sites were reported for tunnel [4, 12, 13] and highway [14]. Although, the concentration levels and workers' exposure to contaminants in tunnel and highway construction have been reported by many researchers, its evaluation for excavation works in top-down constructions is still missing.

Underground excavation workplaces that employ the top-down construction method typically exhibit a working environment similar to mines and tunnel construction sites well known as locations with high concentrations of exposure to DEE [15]. DEE is a composite substance comprised of gaseous substances, including $CO$, $CO_2$, $NO_x$, and VOCs, and particulate matter, such as elemental carbon (EC), organic carbon (OC), sulfate compounds, and polycyclic aromatic hydrocarbons (PAHs) [12, 16]. DEE is classified as a definite material causing lung cancer (Group 1) and as a suspect one causing bladder cancer [17]. In the 1990s, EC was evaluated to be a representative, indicative substance of DEE because, in contrast with organic carbon generated from artificial or natural sources, the dominant source of EC are diesel [15, 18]. Although carbon monoxide, nitrogen monoxide, and nitrogen dioxide have been used as indicators to assess the risks of exposure to DEE in the past, there are limitations in the low specificity and sensitivity [18–21]. Along with recent strengthening of regulations on DEE, the emission of particulate matter, including EC, from DEE has been significantly reduced. Therefore, studies using alternative indicators such as nitrogen dioxide are currently in progress. However, there are problems with doing so since nitrogen oxide can also be contained in the blasting fumes in the excavation sites from the use of blast powder [12].

PAHs are carcinogens in DEE consisting of an aromatic hydrocarbon with more than 2 benzene rings [22, 23]. PAHs, highly soluble in lipids, resulting in a higher residual tendency, bio-concentration, and easy absorption into internal organs through the lungs and skin, known to cause lung, skin, kidney, bladder cancers and reproductive mutation in DNA [24–26]. Underground excavation workers also can be exposed to RCS contained in rocks and soils

in the ground while they are engaging in excavation. RCS is a Group 1 carcinogen, as defined by IARC, and has been reported to cause lung cancer, silicosis, and renal disease. There are no known treatment options other than preventive measures for diseases caused by exposure to silica [27]. However, the problem is that there are no reported data on how high concentrations of crystalline silica, diesel engine exhaust and dust may be exposed to workers working in underground excavation workshops due to insufficient ventilation.

Thus the research questions of present study intends to evaluate the concentration of EC and PAHs (as representative indicators of DEE), RCS, and other respirable particulates to which workers are exposed in underground excavation sites employing top-down construction methods, and also evaluate the correlations between these hazardous substances.

## Materials and methods

### Exposure group selection and task description

Four new residential complex construction sites employing top-down construction methods (Table 1 and S1 Fig) were selected to assess the concentration of EC, OC, TC, dust and crystalline silica in total and respirable particulates and PAHs during underground excavation work between April and May in 2017. Sites with diverse ground conditions were selected, including hard, medium and soft rocks/soils. The construction process rate was 17–20% at the time of evaluation, and four sites were under construction on each of the 2nd and 3rd basements. The excavation work was performed by breaking rocks and transporting soil and rocks to an externally connected opening using diesel-powered engine excavators. The number of excavators used on the day of evaluation varied from 3 to 10 depending on the site situation, and the excavator's diesel engines were manufactured between 2009 and 2016, mostly Euro 5 models. At each of four construction sites, EC and PAHs were assessed for two days at three locations inside and outside the excavator. In addition, at the construction sites B and C, EC, PAHs and total dust, respirable dust and RCS were assessed outside the excavator and at construction site D, total dust, respirable dust and RCS were assessed both inside and outside the excavator for two days. Therefore, 24 EC (each total and respirable), 24 PAHs, and 6 RCS were collected inside the excavator, and 30 EC (each total and respirable), 30 PAHs, and 11 RCS were collected from the outside of the excavator. However, the results were summarized excluding some outlier and missing samples. The excavation work was carried out continuously for more than 8 hours a day, and the sample collection times ranged from 383 to 512 minutes per sample.

**Table 1. Target monitoring workplace.**

| Construction site | A | B | C | D |
|---|---|---|---|---|
| Location (City) | Busan, South Korea | Daegu, South Korea | Daegu, South Korea | Ulsan, South Korea |
| Sampling period | April. 2017 | April. 2017 | April. 2017 | May. 2017 |
| Sampling days | 2 | 3 | 3 | 2 |
| Ground type | Soil | Soft rock | Hard rock | Soil |
| Ventilation type | Half-enclosed | Enclosed | Enclosed | Enclosed |
| Working Ground level | Basement 3 | Basement 2 | Basement 2 | Basement 3 |
| Area (m$^2$) | 7,000 | 3,068 | 8,348 | 3,068 |
| Number of vehicles (Year of Manufacture) | 4 (2009 ~ 2013) | 5 (2010 ~ 2016) | 10 (2012) | 3 (2010 ~ 2015) |
| Area(m$^2$)/ No. vehicle | 1,750 | 613.6 | 834.8 | 1022.7 |
| Blasting work | No | No | Yes | No |

## Sampling and analysis

**Element Carbon (EC), Organic Carbon (OC) and Total Carbon (TC) analysis.** EC, OC and TC were collected on quartz-fiber filters (37-mm in diameter; SKC Inc., USA) mounted on 3-piece cassette and aluminum cyclone (SKC Inc., USA) connected to a pump (Escort Elf Pump; MSA, USA) with a flow rate of 2 L/min for total particulates and 2.5 L/min for respirable particulates. The pumps were pre- and post-calibrated using a dry calibrator (Defender 520-M; MesaLabs, USA). EC, OC and TC were analyzed using OCEC Analyzer (Model 5L; Sunset Lab. Inc., USA) in accordance with the NIOSH Manual of Analytical Methods (NMAM) of the U.S. National Institute of Occupational Safety and Health (NIOSH) #5040 [28]. Three analytical samples were made from the collected quartz-fiber filter and the blank sample, introduced into the analyzer, and the concentration was calculated by multiplying the filter area (8.75 $cm^2$) by the average of the analyzed results for each sample. The analysis conditions of the elemental carbon analyzer are shown in S1 Table, and the detection limits were 0.0008 $\mu$g / sample of EC, 5.0527 $\mu$g / sample of OC, and 5.0527 $\mu$g / sample of TC.

**Polycyclic aromatic hydrocarbons analysis.** PAHs was collected on PTFE filters (Polyetrafluoroethylene, 37-mm in diameter, 2 $\mu$m pores, SKC Inc., USA) and washed XAD-2(100 mg/50 mg, ORBO 43 Supelco; Merck, Germany) connected to a pump (Escort Elf Pump; MSA, USA) with a flow rate of 2 L/min. The pumps were pre- and post-calibrated using a dry calibrator (Defender 520-M; MesaLabs, USA). Samples were wrapped in silver foil during and after sampling to prevent exposure to sunlight (heat and ultraviolet rays), refrigerated, and transported. PAHs was analyzed using a liquid chromatograph (Acquity UPLC H-Class; Waters corp., USA) -fluorescence detector (350 nm / 397 nm) in accordance with NIOSH #5506 [29]. Among the detailed compounds of PAHs, Naphtalene, Acenaphthene, Phenanthrene, Anthracene, Fluoranthene, Pyrene, Benz (a) anthracene, Chrysene, Benzo (b) fluoranthene, Benzo (k) fluoranthene, Benzo (a) pyrene, Fluorene, Acenaphthylene, Debenz (a, h) Concentrations of 16 substances such as anthracene, Benzo (ghi) perylene, and Indeno (1,2,3-C, D) pyrene were evaluated. Benzo (a) pyrene (BaP) equivalent concentration (BaP$_{eq}$) was estimated to find PAH carcinogenic potency relative to BaP. The detection limits and toxic equivalent factor (TEF) for each the detailed compounds of PAHs are shown in S2 Table.

**Gravimetric analysis of dust.** Total and respirable dust were collected on gravimetrically analyzed PVC filters (poly vinyl chloride, 37-mm in diameter, 5 $\mu$m pores; SKC Inc., USA) mounted on 3-piece cassette and aluminum cyclone (SKC Inc, USA) connected to a pump (Escort Elf Pump; MSA, USA) with a flow rate of 2 L/min for total dust and 2.5 L/min for respirable dust. The pumps were pre- and post-calibrated using a dry calibrator (Defender 520-M; MesaLabs, USA). For the gravimetric analysis of dust, PVC filters were dried in a desiccator for over a day before sampling, stabilized in the gravimetric analysis chamber for > 2 hours, and weighed three times using an electronic balance with $10^{-7}$ g readability (XP2U; Mettler Toledo, Switzerland) to calculate the mean value. The samples and blanks were dried, weighed and calculated same as pre-filters.

**Crystalline silica analysis.** After the gravimetric analysis of dust in PVC filters, crystalline silica was analyzed using Fourier-transform infrared spectroscopy (FT-IR) in accordance with the NIOSH #7602 [30]. To pre-treat the samples, the filter was placed in a jar and heated for 2 h in an electrical furnace set at 600°C. Potassium bromide (KBr, 300 mg) (FT-IR grade, Sigma-Aldrich) was added to the jar containing the filter ashes, mixed, and pressed in a 13- mm pellet die to make pellets. FT-IR (Alpha-T; Bruker, Germany) was used to measure the sample's absorbance at 600 to 900 $cm^{-1}$ vibrations and the absorbance at 800 $cm^{-1}$ vibrations was used to calculate the results. The calibration curve was created from 5 to 500 $\mu$g using SRM2950a respirable alpha quartz from the National Institute of Standards and Technology's (NIST) as

the standard. If it was not possible to form pellets because there was too much dust or the amount of quartz exceeded the range of the calibration curve, a portion of the dust was separated to determine the ratio of the dust sample weight to the weight of the total amount of dust. The limit of detection (LOD) was 0.009 mg per sample.

**Statistical analysis.** The results were tested for normality of resources to examine the characteristics of distribution using the Shapiro-Wilk test. Evaluation groups did not follow the normal or log-normal distribution. However, the form of the data is close to the lognormal distribution. The geometric mean (GM), geometric standard deviation (GSD), arithmetic mean (AM), standard deviation (SD) and median were used to explain the concentrations by construction sites. Nonparametric test and spearman correlation analysis were conducted to compare the mean exposure concentration for each construction site and to assess the relationship among EC, OC, CS and dust concentrations. Statistical analyses were performed using PASW version 18.0 (SPSS Inc., Chicago, IL, USA). The figures in this study were generated using Sigma Plot version 14.0 (Systat Software Inc., San Jose, CA, USA).

## Results

### Element carbon, organic carbon and total carbon

Table 2 and Fig 1 shows the EC, OC, and TC concentrations from inside and outside the excavators in underground excavation sites. The GM of respirable EC, OC, and TC concentration were 8.69 $\mu g/m^3$, 34.32 $\mu g/m^3$ and 44.96 $\mu g/m^3$ from inside the excavator and 33.20 $\mu g/m^3$,

**Table 2. Concentration of EC, OC, TC in underground excavation work ($\mu g/m^3$).**

| Classification | | | EC | OC | TC | OC/EC (ratio) |
|---|---|---|---|---|---|---|
| Inside the vehicle | Total particulates | n | 23 | 23 | 23 | 23 |
| | | AM±SD | 13.36±12.02 | 56.24±49.12 | 69.60±57.77 | 5.68±3.75 |
| | | GM(GSD) | 9.57(2.325) | 44.82(1.846) | 55.94(1.856) | 4.68(1.891) |
| | | Median | 9.12 | 34.93 | 50.77 | 5.21 |
| | | Range | 2.09~52.22 | 23.78~192.81 | 26.47~233.66 | 1.35~16.77 |
| | Respirable particulates | n | 23 | 23 | 23 | 23 |
| | | AM±SD | 12.75±13.13 | 44.40±43.84 | 57.15±53.45 | 5.17±4.04 |
| | | GM(GSD) | 8.69(2.425) | 34.32(1.87) | 44.96(1.857) | 3.95(2.136) |
| | | Median | 8.44 | 27.64 | 39.86 | 4.44 |
| | | Range | 1.57~58.43 | 19.11~166.90 | 22.90~225.33 | 0.83~14.95 |
| | Nonparametric_test (p value) | | 0.717 | 0.009 | 0.102 | 0.339 |
| Outside the vehicle | Total particulates | n | 29 | 29 | 29 | 29 |
| | | AM±SD | 50.95±38.09 | 79.59±46.22 | 130.53±79.41 | 2.21±1.28 |
| | | GM(GSD) | 32.02(3.21) | 61.29(2.309) | 96.20(2.516) | 1.91(1.712) |
| | | Median | 53.03 | 91.27 | 159.85 | 1.81 |
| | | Range | 3.24~132.48 | 9.55~171.48 | 13.85~303.97 | 0.82~6.13 |
| | Respirable particulates | n | 30 | 30 | 30 | 30 |
| | | AM±SD | 58.53±49.91 | 51.73±30.89 | 110.27±79.69 | 1.57±1.25 |
| | | GM(GSD) | 33.20(3.615) | 41.53(2.078) | 78.21(2.556) | 1.25(1.887) |
| | | Median | 53.23 | 52.72 | 115.36 | 1.00 |
| | | Range | 3.15~191.42 | 8.33~123.54 | 12.21~314.96 | 0.62~5.28 |
| | Nonparametric_test (p value) | | 0.785 | 0.012 | 0.172 | p<0.05 |
| Nonparametric_test b/w sampling place | | | p<0.0001 | p = 0.069 | p<0.01 | p<0.0001 |

n: Sample number, AM: Arithmetic Mean, SD: Standard deviation, GM: Geometric Mean, GSD: Geometric standard deviation

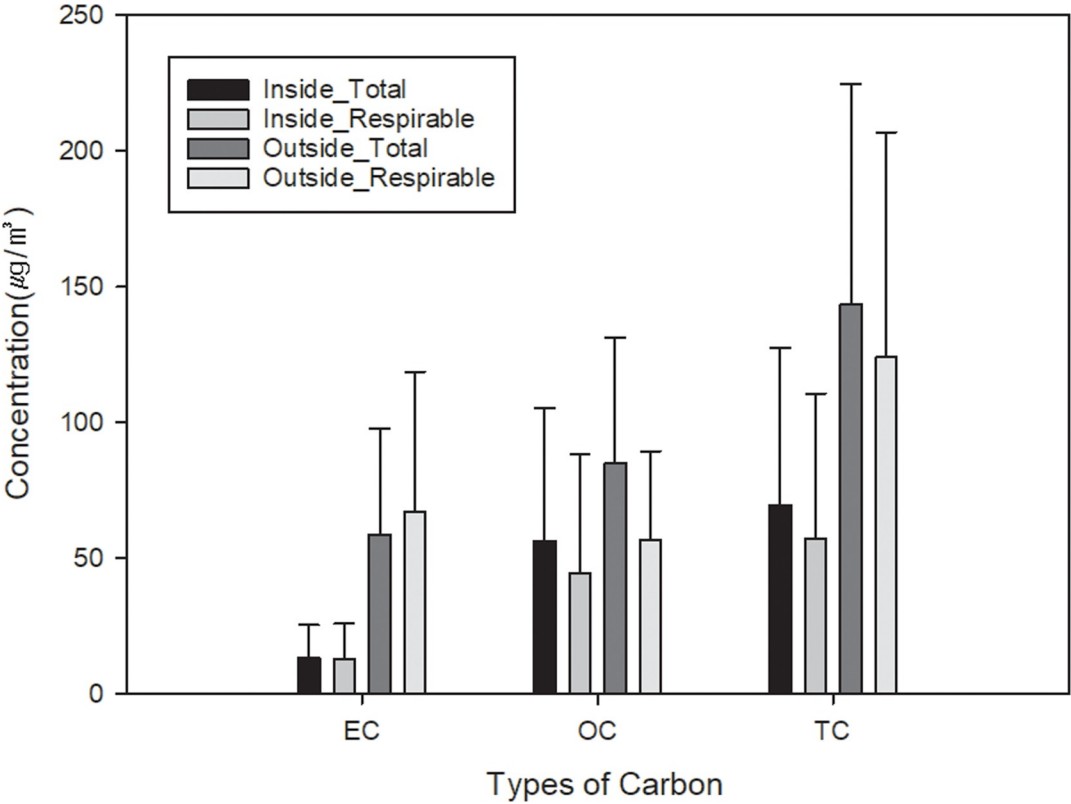

**Fig 1. Mean concentration of EC, OC and TC, along with standard deviation (error bar), for each total and respirable particulates from inside and outside the excavator in underground excavation site.**

41.53 μg/m$^3$ and 78.21 μg/m$^3$ from outside the excavator in underground excavation workshop, respectively. The GM of total EC, OC, and TC concentration were 9.57 μg/m$^3$, 44.82 μg/m$^3$ and 55.94 μg/m$^3$ from inside the excavator and 32.02 μg/m$^3$, 61.29 μg/m$^3$ and 96.20 μg/m$^3$ from outside the excavator in underground excavation workshop, respectively. The EC and TC concentration from outside the excavator is significantly higher than that of inside the excavator (p<0.01). However, OC concentration between inside and outside the excavators is not significantly different. There was no significant difference of EC, OC, and TC concentration between total and respirable particulates.

The OC / EC ratios from inside the excavator was 4.68 (1.35 ~ 16.77) in the total particulates and 3.95 (0.83 ~ 14.95) in the respirable particulates. The OC / EC ratios from outside the excavator in the underground workshop was 1.91 (0.82 ~ 6.33) in the total particulates and 1.25 (0.62 ~ 5.28) in the respirable particulates (Table 2). The concentration of EC by the construction sites (A~D) were significantly different (p < 0.0001). However, there was no significant difference in OC concentration. The OC / EC ratio also differed according to the construction sites (p <0.001) (S3 and S4 Tables and Fig 2).

Table 3 shows the respirable EC concentrations by environmental variables. By ground types, the highest concentration of EC concentration was measured in hard rock ground (27.41 μg/m$^3$), followed by the soft rock ground (9.99 μg/m$^3$), and soil ground (5.02 μg/m$^3$). The EC in enclosed work environment (11.21 μg/m$^3$) was higher than that in half-enclosed work environment (4.23 μg/m$^3$).

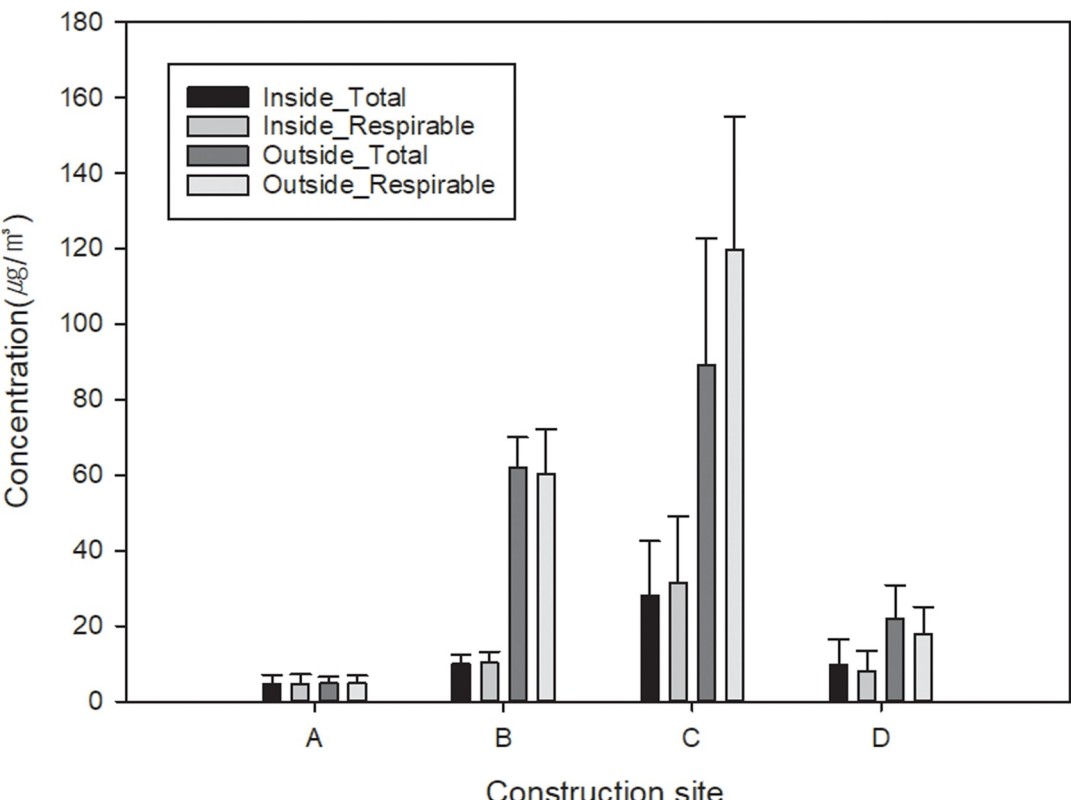

**Fig 2. Mean concentration of EC, along with standard deviation (error bar), for each total and respirable particulates from inside and outside the excavator in underground excavation site.**

## Polycyclic aromatic hydrocarbons

Table 4 and Fig 3 shows the concentration of total PAHs and 16 sub-compounds of PAHs from inside and outside the excavators in underground excavation sites. The concentration was calculated by summing the concentrations detected in the filter and XAD-2 tube. The GM

**Table 3. Concentration of *respirable EC* by environmental variables (unit: μg/m³).**

| | n | Inside the excavator | | | | P value | n | Outside the excavator | | | | P value |
|---|---|---|---|---|---|---|---|---|---|---|---|---|
| | | AM±SD | GM(GSD) | Range | 95% CI | | | AM±SD | GM(GSD) | Range | 95% CI | |
| Ground type | | | | | | | | | | | | |
| Soil | 12 | 6.25±4.47 | 5.02(2) | 1.57~16.4 | 3.41–9.09 | <0.0001 | 12 | 11.29±8.47 | 8.42(2.27) | 3.15~25.36 | 5.91–16.68 | <0.0001 |
| Soft rock | 6 | 10.28±2.82 | 9.99(1.29) | 8.23~14.43 | 7.33–13.24 | | 9 | 60.35±11.66 | 59.33(1.22) | 43.15~79.01 | 51.39–69.3 | |
| Hard rock | 5 | 31.28±17.83 | 27.41(1.78) | 13.87~58.43 | 9.14–53.42 | | 9 | 119.7±35.13 | 115.71(1.31) | 83.66~191.42 | 92.7–146.7 | |
| Ventilation type | | | | | | | | | | | | |
| Half-enclosed | 6 | 4.68±2.47 | 4.23(1.62) | 2.34~9.3 | 2.08–7.27 | <0.1 | 6 | 4.82±1.99 | 4.54(1.44) | 3.15~8.53 | 2.73–6.91 | <0.01 |
| Enclosed | 17 | 15.59±14.21 | 11.21(2.37) | 1.57~58.43 | 8.29–22.9 | | 24 | 71.96±46.89 | 54.6(2.4) | 4.28~191.42 | 52.16–91.76 | |
| Area/No. excavators | | | | | | | | | | | | |
| > 1,000 | 12 | 6.25±4.47 | 5.02(2) | 1.57~16.4 | 3.41–9.09 | <0.05 | 12 | 11.29±8.47 | 8.42(2.27) | 3.15~25.36 | 5.91–16.68 | <0.0001 |
| < 1,000 | 11 | 19.83±15.85 | 15.81(1.95) | 8.23~58.43 | 9.18–30.48 | | 18 | 90.02±39.71 | 82.86(1.51) | 43.15~191.42 | 70.27–109.8 | |
| Blasting | | | | | | | | | | | | |
| No | 18 | 7.6±4.37 | 6.32(1.94) | 1.57~16.4 | 5.43–9.77 | <0.0001 | 21 | 32.32±26.69 | 19.44(3.21) | 3.15~79.01 | 92.7–146.7 | <0.0001 |
| Yes | 5 | 31.28±17.83 | 27.41(1.78) | 13.87~58.43 | 9.14–53.42 | | 9 | 119.7±35.13 | 115.71(1.31) | 83.66~191.42 | 20.17–44.47 | |

n: Sample number, AM: Arithmetic Mean, SD: Standard deviation, GM: Geometric Mean, GSD: Geometric standard deviation, CI: Confidence Interval

**Table 4. Comparison of PAHs concentration between inside and outside of the excavator (unit: μg/m³).**

| PAHS | n | Inside the excavator | | | | n | Outside the excavator | | | |
|---|---|---|---|---|---|---|---|---|---|---|
| | | AM±SD | GM(GSD) | Median | Range | | AM±SD | GM(GSD) | Median | Range |
| Naphthalene | 21 | 1.34±2.01 | 0.67(3.10) | 0.26 | 0.26~8.81 | 29 | 0.62±0.63 | 0.42(2.29) | 0.26 | 0.26~2.22 |
| Acenaphthylene | 21 | 2.10±2.33 | 0.98(4.05) | 1.34 | 0.18~8.53 | 29 | 0.82±0.99 | 0.427(3.077) | 0.18 | 0.18~3.44 |
| Acenaphthene | 21 | 0.76±0.61 | 0.44(3.55) | 0.75 | 0.08~2.17 | 29 | 1.49±1.68 | 0.543(5.48) | 1.02 | 0.08~5.76 |
| Fluorene | 21 | 0.57±0.86 | 0.24(3.59) | 0.23 | 0.07~2.94 | 29 | 0.13±0.19 | 0.091(1.99) | 0.07 | 0.07~0.97 |
| Phenanthrene | 21 | 0.37±0.84 | 0.11(4.44) | 0.096 | 0.012~3.88 | 29 | 0.44±0.97 | 0.078(6.49) | 0.063 | 0.012~4.14 |
| Anthracene | 21 | 0.047±0.042 | 0.032(2.9) | 0.039 | 0.002~0.18 | 29 | 0.04±0.049 | 0.014(5.83) | 0.018 | 0.002~0.14 |
| Fluoranthene | 21 | 0.035±0.04 | 0.024(2.19) | 0.016 | 0.016~0.13 | 29 | 0.22±0.94 | 0.032(3.89) | 0.016 | 0.016~5.08 |
| Pyrene | 21 | 2.47±2.6 | 1.163(4.33) | 1.38 | 0.039~8.34 | 29 | 0.59±0.54 | 0.316(3.84) | 0.57 | 0.039~2.34 |
| Benz(a)anthracene | 21 | 0.005±0.007 | 0.004(1.64) | 0.004 | 0.004~0.037 | 29 | > LOD | > LOD | > LOD | > LOD |
| Chrysene | 21 | > LOD | > LOD | > LOD | > LOD | 29 | > LOD | > LOD | > LOD | > LOD |
| Benzo(b)fluoranthene | 21 | > LOD | > LOD | > LOD | > LOD | 29 | > LOD | > LOD | > LOD | > LOD |
| Benzo(k)fluoranthene | 21 | > LOD | > LOD | > LOD | > LOD | 29 | > LOD | > LOD | > LOD | > LOD |
| Benzo(a)pyrene | 21 | > LOD | > LOD | > LOD | > LOD | 29 | > LOD | > LOD | > LOD | > LOD |
| Debenz(a,h)anthracene | 21 | 0.30±0.18 | 0.28(1.36) | 0.26 | 0.26~1.08 | 29 | > LOD | > LOD | > LOD | > LOD |
| Benzo(ghi)perylene | 21 | > LOD | > LOD | > LOD | > LOD | 29 | > LOD | > LOD | > LOD | > LOD |
| Indeno(1,2,3-C,D)pyrene | 21 | > LOD | > LOD | > LOD | > LOD | 29 | > LOD | > LOD | > LOD | > LOD |
| *Total PAHs* | | *8.34±5.24* | *6.82(1.99)* | *7.31* | *1.26~23.62* | *29* | *4.968±3.619* | *3.93(2.01)* | *3.94* | *1.26~15.34* |

n: Sample number, AM: Arithmetic Mean, SD: Standard deviation, GM: Geometric Mean, GSD: Geometric standard deviation, LOD: Limit of Detection

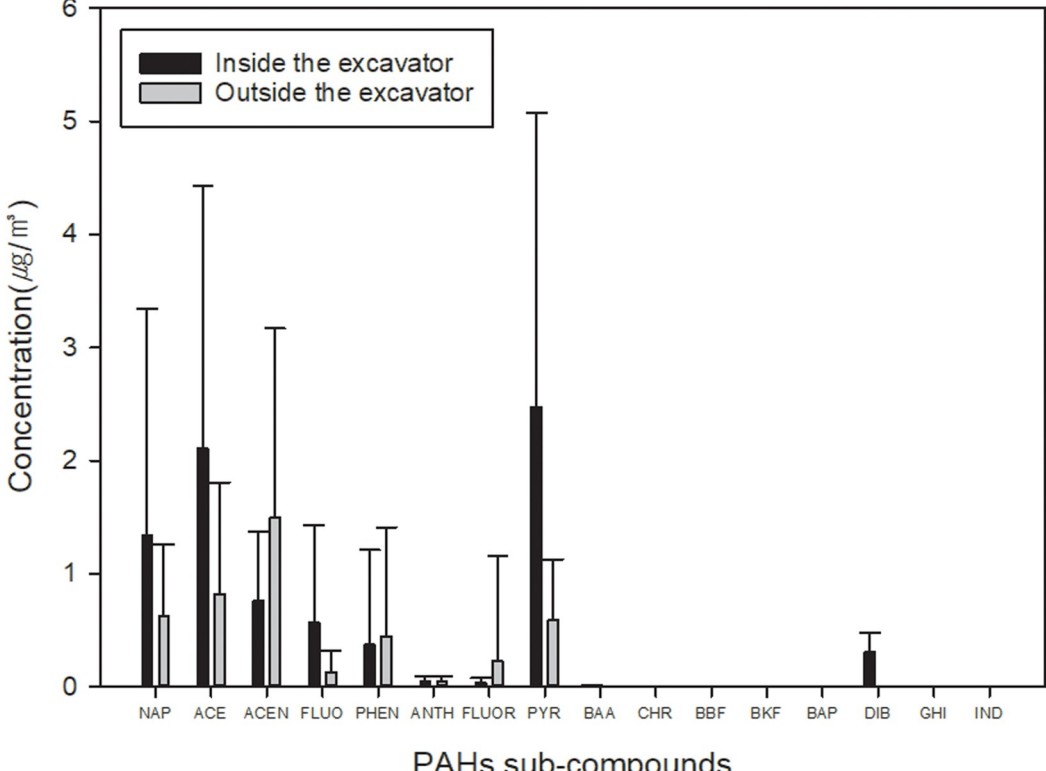

**Fig 3. Mean concentration of PAHs sub-compounds, along with standard deviation (error bar), from inside and outside the excavator in underground excavation site.** NAP: Naphthalene, ACE: Acenaphthylene, ACEN: Acenaphthene, FLUO: Fluorene, PHEN: Phenanthrene, ANTH: Anthracene, FLOUR: Fluoranthene, PYR: Pyrene, BAA: Benz(a) anthracene, CHR: Chrysene, BBF: Benzo(b)fluoanthene, BAP: Benzo(a)pyrene, DIB: Dibenz(a,h)anthracene, IND: Indeno (1,2,3-C,D)pyrene.

of the total PAHs collected from inside the excavator was 6.82 μg/m³ (1.26 ~ 23.62 μg/m³), which was higher than that for outside the excavator in the underground workshop 3.93 μg/m³ (1.26 ~ 15.34 μg/m³). Naphthalene, Acenaphthylene, and Acenaphthene were dominant sub-compounds of PAHs in both inside and outside the excavators, but Pyrene is specifically high inside the excavators. Suspected and animal carcinogens (Benz (a) anthracene, Chrysene, Benzo (b) fluoranthene, Benzo (a) pyrene) were evaluated below detection limits from both in and outside the excavators, however, Benz (a) anthracene was detected inside the excavators as GM 0.004 μg/m³ (0.004 to 0.037 μg/m³). $BaP_{eq}$ was 0.312 μg/m³ and 0.005 μg/m³ from inside and outside the excavator, respectively The PAHs concentration from outside the excavator was affected by environmental variables of ground type, ventilation, vehicle density and blasting ($p<0.05$) (Table 5).

## Total dust, respirable dust and crystalline silica

Table 6 shows dust and crystalline silica concentration from inside and outside the excavators in underground excavation sites. From inside the excavator, the GM of total dust(TD), respirable dust(RD), of total crystalline silica and respirable crystalline silica was 0.24 mg/m³ (0.04 ~ 0.97 mg/m³), 0.13 mg/m³ (0.04 ~ 0.46 mg/m³), 0.025 mg/m³ (0.01 ~ 0.08 mg/m³) and 0.02 mg/m³ (0.01 ~ 0.04 mg/m³). From outside the excavators in the underground workshop, the GM of total dust, respirable dust, of total crystalline silica and respirable crystalline silica was 2.5 mg/m³ (1.13 ~ 4.56 mg/m³), 0.90 mg/m³ (0.46 ~ 1.62 mg/m³), 0.17 mg/m³ (0.07 ~ 0.29 mg/m³) and 0.08 mg/m³ (0.04 ~ 0.15 mg/m³).

## Correlation among EC, OC, TC, dust and crystalline silica concentration

The results of the correlation analysis between EC, OC, TC, crystalline silica, and dust (Table 7 and Fig 4) showed that EC was strongly correlated with OC (r = 0.773, p <0.01) and dust concentrations (r = 0.690, p < 0.01). There was no correlation between the PAHs and EC, but a weak correlation was shown with OC (r = 0.372, p <0.01). Crystalline silica showed a strong correlation with the dust concentration (r = 0.979, p <0.01).

**Table 5. Concentration of *PAHs* by environmental variables (unit: μg/m³).**

| | n | Inside the excavator | | | | *P value* | n | Outside the excavator | | | | *P value* |
|---|---|---|---|---|---|---|---|---|---|---|---|---|
| | | AM±SD | GM(GSD) | Range | 95% CI | | | AM±SD | GM(GSD) | Range | 95% CI | |
| Ground type | | | | | | | | | | | | |
| Soil | 9 | 6.32±6.81 | 4.46(2.3) | 1.262~23.618 | 1.1–11.55 | >0.05 | 12 | 2.63±1.24 | 2.34(1.68) | 1.26~4.23 | 1.85–3.42 | <0.05 |
| Soft rock | 6 | 11.83±3.23 | 11.43(1.35) | 7.133~15.795 | 8.45–15.22 | | 8 | 7.16±5.04 | 5.73(2.05) | 2.30~15.34 | 2.94–11.37 | |
| Hard rock | 6 | 7.86±1.70 | 7.69(1.27) | 5.031~9.416 | 6.08–9.65 | | 9 | 6.13±2.65 | 5.62(1.58) | 2.52~10.93 | 4.1–8.17 | |
| Ventilation type | | | | | | | | | | | | |
| Half-enclosed | 3 | 4.09±1.37 | 3.94(1.39) | 2.906~5.587 | 0.68–7.49 | >0.05 | 6 | 1.92±0.89 | 1.77(1.53) | 1.22~3.44 | 0.98–2.85 | <0.05 |
| Enclosed | 18 | 9.05±5.32 | 7.47(2.01) | 1.262~23.618 | 6.4–11.69 | | 23 | 5.76±3.65 | 4.85(1.83) | 1.26~15.34 | 4.19–7.34 | |
| Area/No. excavators | | | | | | | | | | | | |
| > 1,000 | 9 | 6.32±6.81 | 4.46(2.3) | 1.262~23.618 | 1.1–11.55 | >0.05 | 12 | 2.63±1.24 | 2.34(1.68) | 1.26~4.23 | 1.85–3.42 | <0.05 |
| < 1,000 | 12 | 9.85±3.22 | 9.37(1.39) | 5.031~15.795 | 7.8–11.89 | | 17 | 6.62±3.86 | 5.67(1.78) | 2.30~15.34 | 4.63–8.6 | |
| Blasting | | | | | | | | | | | | |
| No | 15 | 8.53±6.16 | 6.50(2.25) | 1.262~23.618 | 5.11–11.94 | >0.05 | 20 | 4.44±3.93 | 3.35(2.10) | 1.26~15.34 | 2.6–6.28 | >0.05 |
| Yes | 6 | 7.86±1.70 | 7.69(1.27) | 5.031~9.416 | 6.08–9.65 | | 9 | 6.13±2.65 | 5.62(1.58) | 2.52~10.93 | 4.1–8.17 | |

n: Sample number, AM: Arithmetic Mean, SD: Standard deviation, GM: Geometric Mean, GSD: Geometric standard deviation, CI: Confidence Interval

**Table 6. Comparison of dust concentration and crystalline silica concentration in total particulate and respirable particulate in underground excavation work (unit: mg/m³).**

| Classification | Sampling site | Classification | Total particulate | Respirable particulate |
|---|---|---|---|---|
| Dust | Inside the vehicle | n | 6 | 6 |
| | | AM±SD | 0.39±0.36 | 0.18±0.16 |
| | | GM(GSD) | 0.24(3.28) | 0.13(2.57) |
| | | Median | 0.26 | 0.11 |
| | | Range | 0.04~0.97 | 0.04~0.46 |
| | Outside the vehicle | n | 11 | 11 |
| | | AM±SD | 2.74±1.16 | 0.97±0.38 |
| | | GM(GSD) | 2.50(1.6) | 0.90(1.52) |
| | | Median | 2.70 | 1.14 |
| | | Range | 1.13~4.56 | 0.46~1.62 |
| | *Nonparametric_test (p value)* | | *p<0.01* | *p<0.01* |
| Crystalline silica | Inside the vehicle | n | 6 | 6 |
| | | AM±SD | 0.03±0.03 | 0.02±0.01 |
| | | GM(GSD) | 0.03(2.33) | 0.02(1.86) |
| | | Median | 0.03 | 0.02 |
| | | Range | 0.01~0.08 | 0.01~0.04 |
| | Outside the vehicle | n | 11 | 11 |
| | | AM±SD | 0.02±0.09 | 0.09±0.04 |
| | | GM(GSD) | 0.17(1.68) | 0.08(1.61) |
| | | Median | 0.17 | 0.08 |
| | | Range | 0.07~0.29 | 0.04~0.15 |
| | *Nonparametric_test (p value)* | | *p<0.01* | *p<0.01* |

n: Sample number, AM: Arithmetic Mean, SD: Standard deviation, GM: Geometric Mean, GSD: Geometric standard deviation

## Discussion

The purpose of the study, the evaluation of DEE, RCS and respiratory dust in underground excavation sites, was well accomplished by appropriate evaluation methods. The EC and TC concentration from outside the excavator is significantly higher than that of inside the excavator (p<0.01). However, there was no significant difference between total and respirable particulates and it is estimated that approximately 80–95% particulates discharged from diesel engines have sizes below 2.5 μm [31]. The occupational exposure limits for EC concentrations

**Table 7. Spearman's rank correlation of hazardous substances; EC, OC, TC, PAHs, silica and dust.**

| Hazards | n | EC | OC | TC | PAHs | Silica | Dust |
|---|---|---|---|---|---|---|---|
| EC | 105 | 1 | .773** | .901** | 0.087 | .688** | .690** |
| OC | 105 | | 1 | .961** | .312** | .804** | .801** |
| TC | 105 | | | 1 | .258** | .731** | .723** |
| PAHs | 98 | | | | 1 | 0.199 | 0.155 |
| Silica | 33 | | | | | 1 | .979** |
| Dust | 33 | | | | | | 1 |

* p<0.05

** p<0.01

n: Sample number, EC: Element carbon, OC: Organic carbon, TC: EC+OC, PAHs: Polycyclic Aromatic Hydrocarbons

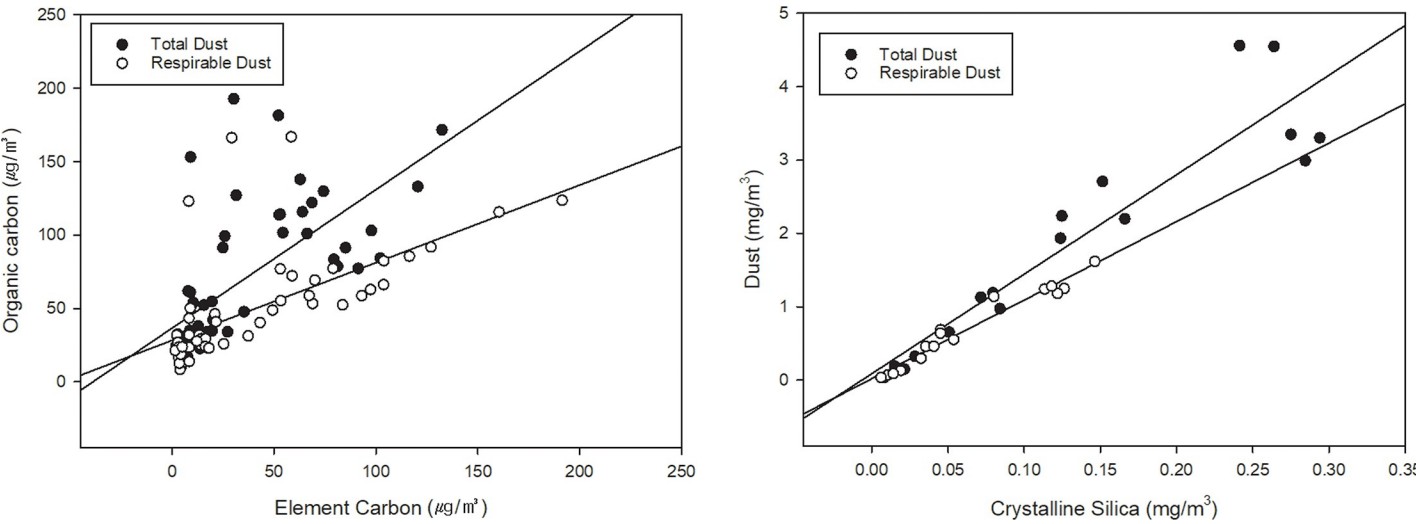

**Fig 4.** Correlation between (a) element carbon and organic carbon, (b) crystalline silica and dust.

is yet to be developed, but recently based on evidence of increased lung cancer at very low levels, the Finnish Institute of Occupational Health (FIOH) recommends the level of standard exposure to EC of 20 µg/m$^3$ among DEE for the underground construction workplaces including mines, and below 5 µg/m$^3$ for other industries, respectively [32]. Council of EU sets an exposure limit of 5 µg/m$^3$ measured in elemental carbon for all diesel engine exhaust fumes [33]. In 2019, the Dutch Expert Committee on Occupational Safety of the Health Council of the Netherlands recommended a health-based OEL for diesel engine exhaust below background levels (approximately 1 µg/m$^3$) [34]. However, the Mine Safety and Health Administration (MSHA) of the US Department of Labor has occupational exposure limits of 160 µg/m$^3$ of concentration of TC (or 120 µg/m$^3$ of elemental carbon) to control the exposure of workers to DEE in each workplace, while the level of standard exposure of 100 µg/m$^3$ of EC has been adopted by Switzerland [35]. In 2001, the American Conference of Governmental Industrial Hygienists (ACGIH®) introduced the 'Notice of Intended Changes (NIC)' of 20 µg/m$^3$ for the concentration of EC and then canceled it later in 2003 [35], because it exceeded the level of chronic exposure to DEE of 5 µg/m$^3$ set by the Office of Environmental Health Hazard Assessment (OEHHA), California, USA [36] by 4 times. In this study, among all EC samples evaluated in the underground excavation workplaces, 9 samples (8.5%) exceeded 100 µg/m$^3$, 34 samples (32.3%) exceeded 50 µg/m$^3$, 49 samples (46.7%) exceeded 20 µg/m$^3$, 83 samples (79.0%) exceeded 5 µg/m$^3$. Thus, pertinent control is needed over excavation equipment, such as excavators or diesel engine vehicles, employed in underground workplaces with insufficient ventilation. The concentration of hazardous elements contained in the exhaust of diesel engines varies according to the type and year of manufacture of the diesel engines, the conditions of the working environment, maintenance of the diesel engines, composition of the fuels, and presence of posttreatment device of exhaust etc. [16, 37], thus the pertinent control and maintenance over replacement or maintenance of old vehicles, use of low sulfur fuel oils, mechanical ventilation of workplaces, and sprinkling of water etc. are needed.

Past data assessing the exposure of workers to EC in underground construction sites were not available. Regarding comparison of the data in this study with those of construction sites of tunnels, the values of GM of exposure to EC were 340 µg/m$^3$ and 100 µg/m$^3$ for workers

engaged in drilling and blasting respectively in a study conducted by [13]. Lewne et al. [19] evaluated the level of EC concentration of 87 μg/m$^3$ assessing in workplaces using diesel engine vehicles for tunnel construction in Stockholm. These appeared to be higher than the level of those identified in this study. Galea et al. [4] evaluated the level of EC concentration in tunnel excavation, and the GM thereof was 18 μg/m$^3$ and highest GM in TBM (tunnel boring machine) tunneling activities was 37 μg/m$^3$ which similar to those obtained in this study. On the contrary, the study, conducted by Hedmer et al. [12] on the exposure of workers to EC in construction sites of railroads tunnels showed that the workers engaged in the operation of tunnel-boring machines had exposure of 2.6 μg/m$^3$ of EC while the whole workplace was exposed to 11 μg/m$^3$ of EC, exhibiting a much lower level than those of the present study. The reason behind the decreased level of exposure to EC, compared to past data, was attributed to a decrease in the creation of particulate matters due to an advancement of recently developed diesel engines.

A comparison with the exposure of workers in different occupations to various levels of concentrations of EC showed similar levels of exposure to firefighters in the United States of America at 35 μg/m$^3$ [38], bus drivers in the United Kingdom at 31 μg/m$^3$ [39], and bus drivers in Estonia at 38 μg/m$^3$ [40]. However, it the exposure was higher than that for workers of concrete pouring at 20 μg/m$^3$, workers of construction sites of express highways at 8 μg/m$^3$, workers of ordinary excavation works at 7 μg/m$^3$ [14, 21], street cleaning workers getting aboard diesel vehicles at 10.7 μg/m$^3$ [21] and 4.8 μg/m$^3$ [41], forklift drivers at 2.1–23.8 μg/m$^3$ [42], workers in the underground parking lots of commercial buildings at 12.2 μg/m$^3$ [43], and of workers engaged in maintenance of buses at 15.5 μg/m$^3$ [44].

Regarding the concentration of EC in different workplaces, the 'Workplace C' exhibited the highest level of concentration of EC wherein 10 excavators were running simultaneously to excavate hard rock ground. This was followed by 'Workplace B' of soft rock ground wherein 5 excavators were running simultaneously. The samples that exhibited a concentration of EC over 100 μg/m$^3$ were all detected at 'Workplace C'. 'Workplace A' and 'Workplace D' with grounds having relatively higher portion of soils had a lower concentration of EC. Therefore, EC concentration varied according to the type of ground, the number of excavators running simultaneously, and the states of ventilation, etc.

The GM of OC/EC ratio in this study from inside the excavator was 3.95 in the respirable particulate, whereas that outside the excavator appeared at 1.25, suggesting the concentration of organic carbon (OC) appeared relatively higher inside the vehicle. In general, it is known that OC/EC ratio ranges from 2 to 3 for the urban area [45]. The OC/EC ratios observed in this study shows that inside the excavator has average higher and underground workshop has average lower than general urban atmosphere.

The concentration of total PAHs was higher inside the excavator than outside the excavator in the underground workplace, contrary to results of the concentrations of EC, RCS and particulates. It is presumed that there was internal pollutants since undetected PAHs sub-compounds from outside the excavator were detected from inside the excavators. However, it was beyond the limitations of this study. The concentration of total PAHs inside the excavator and in the underground workplace were 6.818 μg/m$^3$ and 3.93 μg/m$^3$, respectively, which was lower than the 32.62 μg/m$^3$ of rakers in asphalt pavement sites [46], 17.5 μg/m$^3$ of the paint manufacturing industry [47], 526.55 μg/m$^3$ of the shop of steel pipe coating [47], 10.631 μg/m$^3$ of tar production process in the manufacturing industry of chemical products [48], and higher than 1.884 μg/m$^3$ in vehicle inspection factory in Korea [49] and 0.056 μg/m$^3$ (vehicles of gasoline engine), 0.112 μg/m$^3$ (bus), and 0.199 μg/m$^3$ (vehicles of diesel engine) in vehicle inspection factory in Beijing, China [50]. However, the BaP$_{eq}$ from inside the excavator was similar with that of paint manufacturing and carbon black industry [47, 51] and the BaP$_{eq}$ from

**Table 8. The PAHs concentration reported in earlier studies.**

| Occupational Environments | Process | ∑PAHs (AM. µg/m³) | BaP (µg/m³) | BaPeq (µg/m³) | Country | Reference |
|---|---|---|---|---|---|---|
| Construction industry Underground Excavation | Inside the vehicle | 8.34 | - | 0.312 | S.Korea | This study |
| | Outside the vehicle | 4.97 | - | 0.005 | S.Korea | |
| Asphalt paving | Paver operator | 42.3 | 0.359 | 2.813 | S.Korea | Park et al., 2018 [46] |
| | Raker | 32.618 | 0.267 | 2.071 | | |
| | Macadam roller operator | 7.675 | 0.104 | 0.248 | | |
| | Tire roller operator | 10.792 | - | 0.410 | | |
| Paint Manufacturing | - | 17.48 | 0.3 | 0.394 | S.Korea | Lee, 2004 [47] |
| Steel pipe coating | - | 526.54 | 0.7 | 1.986 | | |
| Tar Manufacturing | Personal | 17.09 | 0.003 | 0.034 | S.Korea | Lee, 2005 [48] |
| | Environmental | 12.97 | 0.11 | 0.46 | | |
| Vehicle Inspection factory | - | 1.884 | 0.007 | 0.017 | S.Korea | Im et al., 2004 [49] |
| Waste Incineration | - | 6.066 | 0.015 | 0.039 | | |
| Vehicle Inspection factory | Bus line | 0.112 | 0.00135 | 0.00438 | China, Beijing | Li et al., 2013 [50] |
| | Gasoline line | 0.0561 | 0.00131 | 0.00334 | | |
| | Diesel line | 0.199 | 0.00434 | 0.0124 | | |
| Carbon black industry | Packaging | 1.953 | 0.341 | 0.566 | Taiwan | Tsai et al., 2001 [51] |
| | Palletizing | 1.449 | 0.285 | 0.314 | | |
| Traffic policeman | Road intersections | 0.867 | 0.0262 | 0.0824 | China, Tianjin | Hu et al, 2007 [52] |
| | Roadsides | 0.0466 | 0.0015 | 0.0057 | | |
| | On Campus | 0.0195 | 0.0007 | 0.0024 | | |
| Highway toll station | - | 0.330 | 0.0216 | 0.0413 | China, Tianjin | Zhao et al., 2016 [53] |

outside the excavator was similar with that in vehicle inspection factory [50] and that of traffic policeman in roadsides [52]. The total PAHs of this study was higher than that of highway toll station [53], but the BaP$_{eq}$ was lower. Table 8 shows the PAHs concentrations reported in earlier studies Bakke et al. [13] evaluated the exposure of 25 workers engaged in drilling, blasting, and concrete work inside of tunnels to PAHs, and reported that all samples appeared to be less than the detection limit ($< 0.2$ µg/m³). Benz(a)anthracene, Chrysene, Benzo(b)fluoranthene, and Benzo(a)pyrene are among 16 sub-compounds of PAHs known to be carcinogens, and they were found to be lower than the detection limit. Thereby, the risk of carcinogenesis was estimated to be low. While the low-molecular PAHs of 3 benzene rings or below are mainly generated from diesel engines, the high molecular PAHs, such as Benzo [a]pyrene and Dibenz [a,h]anthracene etc., are known to be generated from gasoline engines [22, 23]. In contrast, regarding the level of dust concentration in the underground excavation workplaces (employing top-down approach), the GM of total and respirable dust from outside the excavator was 2.498 mg/m³ and 0.901 mg/m³ respectively. which were approximately 10 times higher than the concentration inside the excavator. The GM of 0.0789 mg/m³ of the concentration of RCS among respirable particulates exceeded the occupational exposure limits by approximately 1.5 times, suggesting the presence of risk to catch silicosis or lung cancer. No past assessment on the exposure to RCS concentration in the underground excavation works in construction sites was available. Regard a comparison of construction sites, the exposure of workers drilling, blasting, and conducting concrete works to RCS, as shown in the study conducted by Bakke et al. [13], was reported with geometric mean of 0.025 mg/m³ and 0.033 mg/m³, while Galea et al. [4] reported the exposure of workers engaged in the concrete lining in a tunnel construction site to RCS (in concentration) with a geometric mean of 0.03 mg/m³, and approximately

1/10 of all samples exceeded the level of concentration of RCS of 0.1 mg/m$^3$ in their study. These were similar to the results obtained from the present study. The concentration of TD and RD from outside the excavators was 10 times higher than that of inside the excavators. Brodny and Tutak [54] reported the level of harmful dusts on fully powered longwall coal mines and the research results showed that type of activities and working location had a significant effect on the level of dust exposed. The concentration of RD from outside the excavators was similar with laborer(2.46 mg/m$^3$) and bricklayer (2.13 mg/m$^3$) [9] and much lower than that of painter blaster(13.50 mg/m$^3$) [9], tuck pointer(15.40 mg/m$^3$) [7], recess miller(5.08 mg/m$^3$) and demolition workers(23.67 mg/m$^3$) [8]. Dust showed a strong correlation with the crystalline silica concentration.

## Conclusions

This study is the first assess the exposure of underground excavation worker in top-down construction buildings to EC, PAHs, and CS. The EC and RCS concentration from outside the excavator is significantly higher than that of inside the excavator (p<0.01). The worksite with rock ground, higher vehicle density, blasting and enclosed environments had higher exposure to EC than other sites (p<0.05). EC concentrations during underground excavation work exceeded recommended exposure limits as 20 μg/m$^3$, accounted for about 50% of the total sample, and the level of concentration of RCS exceeded the 1.5 times of occupational exposure limit, 0.05 mg/m$^3$.

Workers working in underground excavation workshops were exposed to high concentrations of crystalline silica, diesel engine exhaust that could have significant adverse health effect. Therefore, engineering and managerial improvement is necessary to improve the working environment in underground excavation workshop. To minimize the effect of exhaust emitted from diesel engine vehicles on the health of the workers in underground excavation sites, filtering devices should be installed at the exhaust outlets of the diesel engines, old vehicles should be replaced, vehicles should be subjected to examination and maintenance, and low sulfur oil should be used. Further, introducing watering facilities to create wet environments in the underground workplaces and a supply of fresh air and ventilation are strongly suggested. This study was aimed to improve the working environment and prevent occupational diseases in underground excavation workers: The hazards are evaluated and the data might be useful for preparing safe engineering and managerial measures.

## Supporting information

**S1 Fig.** Target monitoring workplace (A~D).
(DOCX)

**S1 Table. Analysis condition for OCEC analyzer.**
(DOCX)

**S2 Table. Detection limits of PAHs species.**
(DOCX)

**S3 Table. Concentration of *Total* EC, OC, TC by construction site.**
(DOCX)

**S4 Table. Concentration of *respirable* EC, OC, TC by construction site.**
(DOCX)

## Author Contributions

**Conceptualization:** Hyunhee Park, Chungsik Yoon.

**Data curation:** Eunsong Hwang, Miyeon Jang.

**Funding acquisition:** Hyunhee Park.

**Investigation:** Hyunhee Park, Eunsong Hwang.

**Methodology:** Eunsong Hwang, Miyeon Jang.

**Project administration:** Hyunhee Park.

**Supervision:** Hyunhee Park, Chungsik Yoon.

**Validation:** Eunsong Hwang.

**Writing – original draft:** Hyunhee Park.

**Writing – review & editing:** Chungsik Yoon.

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
