## [Decision Letter · Decision Letter 0]

20 Jul 2020

PONE-D-20-13027

Exposure Assessment of Elemental Carbon, Polycyclic Aromatic Hydrocarbons and Crystalline Silica at the Underground Excavation Sites for Top-Down Construction Buildings

PLOS ONE

Dear Dr. Yoon,

Thank you for submitting your manuscript to PLOS ONE. After careful consideration, we feel that it has merit but does not fully meet PLOS ONE’s publication criteria as it currently stands. Therefore, we invite you to submit a revised version of the manuscript that addresses the points raised during the review process.

In your revised manuscript, kindly pay attention to the following aspects:

You have divided your abstract into four subheadings, i.e. objective, method, result and conclusions. This might confuse the readers. There is no need to give four subheadings in the abstract. It is advised to combine all four sections under one heading “abstract” without any subheading.In the introduction section, kindly describe some previous similar literature and highlight the novelty of your work in relation to the previous studies.Highlight the significance of your outcomes in stronger way in discussion and conclusion parts.Kindly avoid the use of First Person Singular and Plural (I, we) throughout your text.

We look forward to receiving your revised manuscript.

Kind regards,

Anwar Khitab

Academic Editor

PLOS ONE

Journal Requirements:

2.We suggest you thoroughly copyedit your manuscript for language usage, spelling, and grammar. If you do not know anyone who can help you do this, you may wish to consider employing a professional scientific editing service.  

4. We note you have included a table to which you do not refer in the text of your manuscript. Please ensure that you refer to Table 8 in your text; if accepted, production will need this reference to link the reader to the Table

Reviewers' comments:

Reviewer's Responses to Questions

**Comments to the Author**

1. Is the manuscript technically sound, and do the data support the conclusions?

Reviewer #1: Partly

Reviewer #2: Yes

2. Has the statistical analysis been performed appropriately and rigorously? 

Reviewer #1: Yes

Reviewer #2: Yes

3. Have the authors made all data underlying the findings in their manuscript fully available?

Reviewer #1: Yes

Reviewer #2: Yes

4. Is the manuscript presented in an intelligible fashion and written in standard English?

Reviewer #1: Yes

Reviewer #2: Yes

5. Review Comments to the Author

Reviewer #1: The paper deals with a subject that falls within the scope of the journal, but the manuscript needs big improvements concerning the overall presentation. Overall, the article should be written more clearly..

1. There is no abstract in the work. This is unacceptable.

2. The introduction section is poorly organized, it is good to express the need for the study with the backlog of literature exists in the framework. There is no background in this introduction stating the urge and novelty of the study in which innovative ideas must be flown through the background along with the useful insights The introduction must be improved. In introduction, please review the previous studies. What is the novelty for this article compared with existing studies?

3. The problem of dusting occurs in many industries. You also have to write about it, e.g. https://doi.org/10.3390/ijerph15091846

4. The structure of the paper should include the following sections: introduction, theoretical framework, materials method, results, discussion and conclusions.

5. Need strong comment on scientific outcomes.

Reviewer #2: The comments of this reviewer are as follows:

I think the title accurately reflects the purpose of the paper.

The abstract is concise and informative in concerning to matter under investigation, research method adopted and main results and conclusions.

The paper achieves its declared purpose and was well structured and thought out logically.

The English and syntax are correct.

In general, it is possible to state that the manuscript is technically sound, and do the data support the conclusions, and also authors made all data underlying the findings in their manuscript fully available.

The methodology and main research options are summarily referred in abstract and little explored in introduction section. A greater emphasis must be given this matter.

The general research method seems to have been rigorous, clear and properly accounted for. However, some options adopted in concerning to research method and resources used are not adequately explained and defended. For example, options related to statistical analysis should be discussed; and Sampling is not sufficiently clear and defended.

The criteria considered for the selection of the projects (construction sites) are not explained neither discussed. The author(s) should improve and clarify significantly this point.

The references are up to date and represent adequately the previous work and the paper addresses the key issues encompassed by the topic.

Tables and figures are elucidative and help to understand. However all figures should include the axes titles as well as the units in axes. .

Don’t use the first verbal person. Please remove "We", or "Our" throughout the ALL text.

In conclusions or discussion sections, I think it would be better to include a small discussion / analysis on whether the results and findings of this paper corresponded to the objectives of the study and will contribute to the increase in knowledge in the studied domain and to the better performance of industry. What were the main contributions from this study? And what is the use of the findings for future studies?

6. PLOS authors have the option to publish the peer review history of their article (what does this mean?). If published, this will include your full peer review and any attached files.

Reviewer #1: No

Reviewer #2: No

---

## [Author Response · Author response to Decision Letter 0]

18 Aug 2020

# Reviewer 1

1) Comment: There is no abstract in the work. This is unacceptable.

Section: Abstract

Response: When we first submitted the manuscript, we also submitted the abstract, but I wonder why there wasn’t. Anyway, the first abstract had divided into four subheadings before, i.e. objective, method, result and conclusions. It is revised to combine all four sections under one heading “abstract” without any subheading. Please see below

2) Comment: The introduction section is poorly organized, it is good to express the need for the study with the backlog of literature exists in the framework. There is no background in this introduction stating the urge and novelty of the study in which innovative ideas must be flown through the background along with the useful insights The introduction must be improved. In introduction, please review the previous studies. What is the novelty for this article compared with existing studies?

Section: Introduction

Response: In the introduction section, some previous similar literature were described and the novelty of our work was highlighted in relation to the previous studies

3) Comment: The problem of dusting occurs in many industries. You also have to write about it, e.g. https://doi.org/10.3390/ijerph15091846

Section: Discussion

Response: The results of dust exposure in this study were compared with the results of dust exposure in other studies by various types of construction sites.

4) Comment: The structure of the paper should include the following sections: introduction, theoretical framework, materials method, results, discussion and conclusions.

Section: Introduction

Response: 

The theoretical framework was not an essential component in the manuscript organization of PLOS ONE, so the contents were included in the introduction

5) Comment: Need strong comment on scientific outcomes

Section: Discussion and Conclusions

Response: The significance of our outcomes highlighted in conclusion parts

# Reviewer 2

1) Comment: The general research method seems to have been rigorous, clear and properly accounted for. However, some options adopted in concerning to research method and resources used are not adequately explained and defended. For example, options related to statistical analysis should be discussed; and Sampling is not sufficiently clear and defended.

Section: Materials and methods 

Response: A description of the number of samples collected and the statistical analysis method was added.

2) Comment: The criteria considered for the selection of the projects (construction sites) are not explained neither discussed. The author(s) should improve and clarify significantly this point.

Section: Materials and methods

Response: 

The background of the selection of the research site was described. 

3) Comment: The references are up to date and represent adequately the previous work and the paper addresses the key issues encompassed by the topic.

Tables and figures are elucidative and help to understand. However all figures should include the axes titles as well as the units in axes. .

Section: Results

Response: We agree with the reviwer’s opinion 

4) Comment: Don’t use the first verbal person. Please remove "We", or "Our" throughout the ALL text

Section: Abstract

Response: ‘We’ was removed

5) Comment: In conclusions or discussion sections, I think it would be better to include a small discussion / analysis on whether the results and findings of this paper corresponded to the objectives of the study and will contribute to the increase in knowledge in the studied domain and to the better performance of industry. What were the main contributions from this study? And what is the use of the findings for future studies?

Section: Conclusions

Response: The sentence which described the main contributions from this study and further studies is added

---

## [Editor Report · Decision Letter 1]

25 Aug 2020

PONE-D-20-13027R1

Exposure Assessment of Elemental Carbon, Polycyclic Aromatic Hydrocarbons and Crystalline Silica at the Underground Excavation Sites for Top-Down Construction Buildings

PLOS ONE

Dear Dr. Yoon,

Thank you for submitting your manuscript to PLOS ONE. After careful consideration, we feel that it has merit but does not fully meet PLOS ONE’s publication criteria as it currently stands. Therefore, we invite you to submit a revised version of the manuscript that addresses the points raised during the review process.

Although the main concerns of the reviewers have been addressed. These include:

1. Description of Novelty

2. Significance of the work in conclusions and discussion.

However, it is advised to improve the following important linguistic issues:

Abstract:

Remove “the” before “1.5 times” at line 45 and remove the sentence “The concentration of EC, OC, TC, PAHs , dust and CS was assessed for underground excavation worker” (under sub-headings, this sentence works but not in a combined abstract) at lines 48-49. This has already been mentioned.

Introduction:

You have now mentioned the novelty of the work, but the novelty as expressed at line 69 needs to be expressed in a better linguistic manner, like, “Although, the concentration levels and workers’ exposure to contaminants/pollutants in tunnel and highway construction have been reported by many researchers [Ref….], its evaluation for excavation works in top-down constructions is still missing”.

Line 113-115: Only mention that the “Sites with diverse ground conditions were selected, including hard, medium and soft rocks/soils.”

What do you mean by “S2, S3 and … Tables”?

Conclusions

Line 454-456: Complete the sentence. It is linguistically incomplete with no punctuation at the end of the sentence. “Therefore, engineering and managerial improvement necessary to improve the working environment in underground excavation workshop”.

Line 461-464: Kindly change the sentence “Through this study, to improve the working environment and prevent occupational diseases in underground excavation workers, the results of the evaluation of the hazards were derived, and this data might be used for preparing engineering and managerial improvement measures. “ To

“This study was aimed to improve the working environment and prevent occupational diseases in underground excavation workers: The hazards are evaluated and the data might be useful for preparing safe engineering and managerial measures.”

We look forward to receiving your revised manuscript.

Kind regards,

Anwar Khitab

Academic Editor

PLOS ONE

---

## [Author Response · Author response to Decision Letter 1]

26 Aug 2020

Request 1)

Comment: Remove “the” before “1.5 times” at line 45 and remove the sentence “The concentration of EC, OC, TC, PAHs , dust and CS was assessed for underground excavation worker” (under sub-headings, this sentence works but not in a combined abstract) at lines 48-49. This has already been mentioned.

Introduction.

Section: Abstract

Response: “the” before “1.5 times” and the sentence “The concentration of EC, OC, TC, PAHs , dust and CS was assessed for underground excavation worker” was removed

Request 2)

Comment: You have now mentioned the novelty of the work, but the novelty as expressed at line 69 needs to be expressed in a better linguistic manner, like, “Although, the concentration levels and workers’ exposure to contaminants/pollutants in tunnel and highway construction have been reported by many researchers [Ref….], its evaluation for excavation works in top-down constructions is still missing”. 

Section: Introduction

Response: The sentence was expressed in a better linguistic manner as recommended

Request 3)

Comment: Line 113-115: Only mention that the “Sites with diverse ground conditions were selected, including hard, medium and soft rocks/soils.” 

Section: Materials and methods

Response: The sentence was expressed in a better linguistic manner as recommended.

Request 4) 

Comment: What do you mean by “S2, S3 and … Tables”?

Section: Materials and methods

Response: 

S2, S3 Tables are Supporting Information.

“S4 and S5 Tables” was changed by format citations

Request 5)

Comment: Line 454-456: Complete the sentence. It is linguistically incomplete with no punctuation at the end of the sentence. “Therefore, engineering and managerial improvement necessary to improve the working environment in underground excavation workshop”. 

Section: Conclusions

Response: The sentence was completed as recommended.

Request 6)

Comment: Line 461-464: Kindly change the sentence “Through this study, to improve the working environment and prevent occupational diseases in underground excavation workers, the results of the evaluation of the hazards were derived, and this data might be used for preparing engineering and managerial improvement measures. “ To

“This study was aimed to improve the working environment and prevent occupational diseases in underground excavation workers: The hazards are evaluated and the data might be useful for preparing safe engineering and managerial measures.”

Section: Conclusions

Response: The sentence was expressed in a better linguistic manner as recommended.

---

## [Editor Report · Decision Letter 2]

28 Aug 2020

Exposure Assessment of Elemental Carbon, Polycyclic Aromatic Hydrocarbons and Crystalline Silica at the Underground Excavation Sites for Top-Down Construction Buildings

PONE-D-20-13027R2

Dear Dr. Yoon,

We’re pleased to inform you that your manuscript has been judged scientifically suitable for publication and will be formally accepted for publication once it meets all outstanding technical requirements.

Kind regards,

Anwar Khitab

Academic Editor

PLOS ONE
---

## [Editor Report · Acceptance letter]

1 Sep 2020

PONE-D-20-13027R2 

Exposure Assessment of Elemental Carbon, Polycyclic Aromatic Hydrocarbons and Crystalline Silica at the Underground Excavation Sites for Top-Down Construction Buildings 

Dear Dr. Yoon:

I'm pleased to inform you that your manuscript has been deemed suitable for publication in PLOS ONE. Congratulations! Your manuscript is now with our production department. 

Kind regards, 

on behalf of

Dr. Anwar Khitab 

Academic Editor

PLOS ONE